# SlimSAM:
# 0.1% Data Makes Segment Anything Slim

**Zigeng Chen, Gongfan Fang, Xinyin Ma, Xinchao Wang**[*]
National University of Singapore
zigeng99@u.nus.edu, xinchao@nus.edu.sg

## Abstract

Current approaches for compressing the Segment Anything Model (SAM) yield commendable results, yet necessitate extensive data to train a new network from scratch. Employing conventional pruning techniques can remarkably reduce data requirements but would suffer from a degradation in performance. To address this challenging trade-off, we introduce SlimSAM, a novel data-efficient SAM compression method that achieves superior performance with extremely less training data. The essence of SlimSAM is encapsulated in the alternate slimming framework which effectively enhances knowledge inheritance under severely limited training data availability and exceptional pruning ratio. Diverging from prior techniques, our framework progressively compresses the model by alternately pruning and distilling distinct, decoupled sub-structures. Disturbed Taylor pruning is also proposed to address the misalignment between the pruning objective and training target, thereby boosting the post-distillation after pruning. Slim-SAM yields significant performance improvements while demanding **over 10 times less** training data than any other existing compression methods. Even when compared to the original SAM, SlimSAM achieves approaching performance while reducing parameter counts to merely **1.4% (9.1M)**, MACs to **0.8% (23G)**, and requiring only **0.1% (10k)** of the SAM training data. Code is available at https://github.com/czg1225/SlimSAM

## 1 Introduction

*Segment Anything Model* (SAM) [25] has attracted considerable attention from the community since its inception. A plethora of studies [48, 19, 34, 43, 31, 2, 59, 55, 18, 53, 35] have achieved substantial progress by incorporating SAM as a fundamental component. Nevertheless, despite its remarkable performance, SAM's substantial model size and high computational demands render it inadequate for practical applications on resource-constrained devices. This limitation consequently hinders the advancement and broader application of SAM-based models.

To mitigate these constraints, many efforts [29, 62, 60, 50, 63, 45] have been made to effectively compress SAM. Without exception, these endeavors opt to replace the originally heavyweight image encoder with a lightweight and efficient architecture. This invariably entails training a new network from scratch. With regard to scratch training, an unavoidable challenging trade-off arises between training costs and model performance. Existing methods all inevitably compromise performance when training with very limited data.

The crux of the above issue is their inability to fully exploit the capability of pre-trained SAM. To overcome the high training data demands by reusing the robust prior knowledge of pre-trained SAM, a straightforward strategy involves the application of pruning techniques [38, 14, 3, 52, 8] to directly

---

[*]Correspoding Author

38th Conference on Neural Information Processing Systems (NeurIPS 2024).

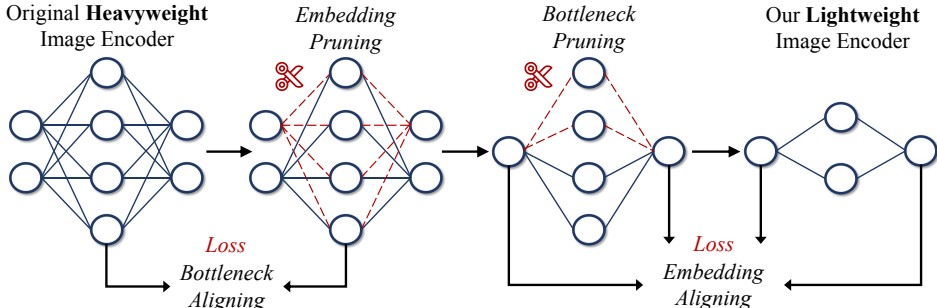

Figure 1: A simple overall diagram of the proposed alternate slimming process.

compress the sizable SAM by removing redundant parameters from the network and fine-tuning the streamlined model with a minimal dataset [16, 36, 10, 30]. Nevertheless, following this conventional procedure leads to unexpected steep performance degradation, particularly when the pruning ratio is set aggressively high and the available data is extremely scarce.

In response to the challenges outlined above, we present SlimSAM, a data-efficient method for SAM compression. Initiating with a standard pruning-finetuning workflow, we gradually "modernize" the compression procedure by introducing our novel designs customized for severely limited data availability and the intricate coupled structure of SAM, culminating in exceptional efficacy while requiring minimal training data. Central to the method are our pioneering contributions: the alternate slimming framework and the disturbed Taylor pruning.

The alternate slimming framework, presented in Figure 1, boosts performance by minimizing divergence from the original model and enabling the intermediate feature alignment via consistent dimensionality. Diverging from prior methods, it alternates between pruning and distillation within decoupled model components. The process begins by targeting the embedding dimensions for pruning and aligning the consistent bottleneck dimensions for distillation. It then shifts focus to pruning the bottleneck dimensions in ViTs [6], aligning the unchanged embedding dimensions for distillation. Observing the misalignment between the pruning object and the distillation target impedes the efficacy of compression, we introduce a novel label-free importance estimation criterion called disturbed Taylor importance to address this misalignment effectively, thereby enhancing the recovery process and obviating the need for labeled data.

Comprehensive assessments across performance metrics, efficiency, and training data requirements reveal that SlimSAM markedly enhances compression performance, concurrently achieving superior lightweight and efficiency with markedly reduced training data requirements. Notably, our entire compression can be completed using only 10k un-labeled images on a single Titan RTX GPU.

In summary, our contribution is a data-efficient SAM compression method called SlimSAM, which effectively repurposes pre-trained SAMs without the necessity for extensive retraining. This is achieved through a novel modernized pruning-distillation procedure. By proposing the alternate slimming framework and introducing the concept of disturbed Taylor importance, we realize greatly enhanced knowledge retention in data-limited situations. When compared to the original SAM-H, SlimSAM achieves approaching performance while reducing the parameter counts to **1.4% (9.1M)**, MACs to **0.8% (23G)**, and requiring mere **0.1% (10k)** of the training data. Extensive experiments demonstrate that our method realizes significant superior performance while utilizing **over 10 times less** training data when compared to any other compression methods.

## 2   Related Works

**Model Pruning**. Due to the inherent parameter redundancy in deep neural networks [13], model pruning [16, 14, 30, 3, 27, 33, 52, 36, 10] has proved to be an effective approach for accelerating and compressing models. Pruning techniques can be generally classified into two main categories: structural pruning [30, 27, 56, 8, 4, 56, 7, 9] and unstructured pruning [5, 26, 40, 42, 12]. Structural pruning is focused on eliminating parameter groups based on predefined criteria, while unstructured pruning involves the removal of individual weights, typically requiring hardware support.

**Efficient Learning**. *Efficient Learning* refers to a range of techniques [54, 57, 20, 21, 58, 11, 22, 28] aimed at reducing the training costs of deep models while maintaining performance. *Knowledge Distillation* (KD) [17] is a prominent method under this category, where knowledge is transferred from a larger, powerful teacher model to a smaller, more efficient student model. This approach leverages soft targets and a temperature parameter to enable the student model to learn more effectively. KD [46, 44, 61, 1, 51, 32, 47, 37, 39] has proven to be an effective strategy for model compression, making it highly applicable in scenarios requiring resource-efficient deployment.

**SAM Compression**. The formidable model size and computational complexity of SAM pose challenges for edge deployment, prompting an extensive array of research focused on devising compression techniques for SAM to enhance its applicability. Notably, FastSAM [62] replaces SAM's extensive ViT-based architecture with the efficient CNN-based YOLOv8-seg [23] model, while MobileSAM [60] adopts the lightweight Tinyvit [49] to replace the image encoder and employs knowledge distillation from the original encoder. EdgeSAM [63] introduces the prompt-in-the-loop knowledge distillation to accurately capture the intricate dynamics between user input and mask generation. EfficientSAM [50] innovatively adapts MAE [15] framework to obtain efficient image encoders for segment anything model but requires extensive training data even more than the SA-1B dataset. However, the above approaches all inevitably suffer from scratch training, resulting in unsatisfactory performance when training data is limited.

**Remark.** The application of common pruning and KD methods falls short in achieving superior performance due to the unique challenges presented by limited training data and SAM's coupled structure. To enhance performance, we propose an alternate slimming framework to minimize divergence from the original model and enable the intermediate feature alignment by consistent dimensionality. We also propose disturbed Taylor pruning to address the misalignment between pruning objectives and training targets. In contrast to other SAM compression methods, our SlimSAM achieves superior compression performance while significantly incurring lower training data requirements.

# 3 Methods

Our paramount objective is to achieve substantial compression of the large image encoder while minimizing performance degradation in scenarios characterized by severe data limitations. To navigate the challenging trade-off between maintaining remarkable performance and the necessity for copious training data, we adopt a strategy of directly inheriting the core weights from the original SAM. This approach capitalizes on SAM's robust prior knowledge, derived from 11 million images. Adhering to this foundational principle, we begin with a standard workflow: initial pruning of the model followed by refinement through post-distillation.

## 3.1 Identifying SAM Redundancy

The initial phase is dedicated to the estimation of the importance of each parameter, determining the non-essential and redundant parameters of the image encoder to be pruned. To fulfill this objective, we endeavor to estimate the importance of a parameter through the quantification of prediction errors engendered by its removal [38]. Given a labeled dataset with N image pairs $\{x_i, y_i\}_{i=1}^{N}$ and a model $\mathcal{F}$ with M parameters $W = \{wi\}_{i=1}^{M}$. The output of the original model can be defined as $t_i = \mathcal{F}_W(x_i)$. Our objective is to identify the parameters that yield the minimum deviation in the loss. Specifically, the importance of a parameter $w_i$, can be defined as:

$$I_{w_i} = |\Delta\mathcal{L}(x_i, y_i)| = |\mathcal{L}_{w_i}(x_i, y_i) - \mathcal{L}_{w_i=0}(x_i, y_i)|, \tag{1}$$

where $\mathcal{L}(x_i, y_i)$ is the loss between the model output and the label $y_i$ when input data is $x_i$. We can approximate $\mathcal{L}_{w_i=0}$ in the vicinity of $w_i$ by its first-order Taylor expansion:

$$\mathcal{L}_{w_i=0}(x_i, y_i) = \mathcal{L}_{w_i}(x_i, y_i) - \frac{\partial\mathcal{L}(x_i, y_i)}{\partial w_i}w_i + \mathcal{R}_1(w_i = 0). \tag{2}$$

Substituting equation 2 into equation 1, we can approximate the parameter importance as:

$$I_{w_i} \approx \left|\mathcal{L}_{w_i=0}(x_i, y_i) - \mathcal{L}_{w_i=0}(x_i, y_i) + \frac{\partial\mathcal{L}(x_i, y_i)}{\partial w_i}w_i\right| = \left|\frac{\partial\mathcal{L}(x_i, y_i)}{\partial w_i}w_i\right|. \tag{3}$$

However, there exist two distinct limitations associated with the above Taylor importance estimation when pruning the image encoder of SAM. Firstly, the accuracy of Taylor importance relies heavily on the availability of sufficiently accurate hard labels $y_i$. Unfortunately, due to the intricate nature of jointly optimizing the image encoder and combined decoder [60], the post-distillation process necessitates performing on the image embedding $t_i$, resulting in the utilization of soft labels exclusively. Secondly, a concern arises regarding the consistency of loss functions when employing Taylor importance estimation for SAM pruning. The importance estimation strategy's primary objective is to identify parameters $w_i$ that minimize the hard label discrepancy $|\Delta\mathcal{L}(x_i, y_i)|$. In contrast, the goal of the distillation-based recovery process is to minimize the soft label loss $|\Delta\mathcal{L}(x_i, t_i)|$. This misalignment in optimization objectives potentially impedes the efficacy of the distillation process. The experimental results in Section 5 also strongly prove our conclusion.

**Disturbed Taylor importance.** To address the unique limitations associated with Taylor importance estimation, we introduce an extremely simple yet effective solution known as disturbed Taylor importance. Given the absence of hard labels and the incongruity of loss functions, a logical approach is to identify parameters $w_i$ that minimize the soft label divergence $|\Delta\mathcal{L}(x_i, t_i)|$. However, the gradients $\frac{\partial\mathcal{L}(x_i,t_i)}{\partial w_i}$ resulting from applying the loss between encoder's outputs $t_i$ are consistently zero. Consequently, we calculate gradients based on the loss function between the original image embedding $t_i$ and disturbed image embedding $t_i + \mathcal{N}(\mu, \sigma^2)$, where $\mathcal{N}$ is Gaussian noise with mean $\mu = 0$ and standard deviation $\sigma = 0.01$. As the expectation $E(t_i + \mathcal{N}) = t_i$, when the batch size is large enough, the importance of a parameter $w_i$ can be approximated as:

$$
\begin{aligned}
I_{w_i} = |\Delta\mathcal{L}(x_i, t_i)| &\approx |\Delta\mathcal{L}(x_i, t_i + \mathcal{N})| \\
&= |\mathcal{L}_{w_i}(x_i, t_i + \mathcal{N}) - \mathcal{L}_{w_i=0}(x_i, t_i + \mathcal{N})| \\
&\approx \left|\frac{\partial\mathcal{L}(x_i, t_i + \mathcal{N})}{\partial w_i} w_i\right|.
\end{aligned}
\tag{4}
$$

As the generated gradients $\frac{\partial\mathcal{L}(x_i, t_i+\mathcal{N})}{\partial w_i} \neq 0$, the importance can be estimated.

**Remark.** Leveraging our disturbed Taylor importance, the pruning objective is seamlessly aligned with the optimization target of subsequent distillation. Compared to previous pruning techniques, it results in a 0.85% MIoU enhancement when the pruning ratio reaches 77% and a 0.60% MIoU improvement when the pruning ratio is set at 50%. Moreover, the adoption of disturbed Taylor importance transforms the entire compression workflow into a convenient label-free framework without incurring additional computational costs.

### 3.2 Alternate Slimming.

After estimating the weights' importance, our approach advances to implementing channel-wise structural pruning on the extensive image encoder, followed by distillation-based model finetuning. To attain an unprecedentedly high compression rate, the pruning ratio in this study is necessitated to be set significantly higher than in typical scenarios. With the pruning ratio exceeding 75%, we observe a marked performance degradation between the pruned model and its original counterpart, a consequence of employing the conventional single-step pruning technique. Additionally, the extremely constrained data availability also poses unique challenges to distillation efficacy. Employing merely 0.1% of the SA-1B dataset (10k images) for post-distillation underscores a significant challenge in recuperating satisfactory performance for the pruned model.

To address identified challenges, we introduce an innovative alternate slimming framework, anchored by two principles: reducing the divergence between the original and pruned models, and enhancing post-distillation efficacy.

Our framework decomposes the model into two separate sub-structures: embedding (output dimensions of each block) and bottleneck (intermediate features of each block). By sequentially pruning and restoring either sub-structure, we achieve a smoother compression loss, preventing the steep performance degradation typically associated with extreme pruning ratios. To improve post-distillation, we exploit the hidden state information of the original model. Due to the structural resemblance between the pruned and original models, using intermediate hidden states for supervision facilitates superior knowledge transfer. Traditional pruning workflow struggles with dimensionality inconsistency, complicating hidden state supervision. Our method, by partitioning the model into

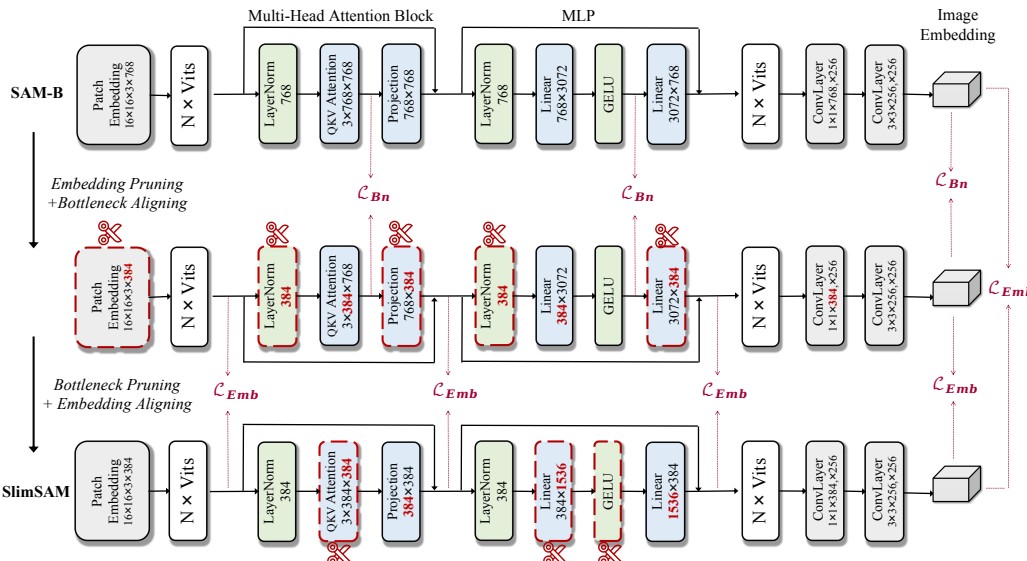

Figure 2: The provided figure depicts our alternate slimming process with a 50% pruning ratio on SAM-B. We utilize structural pruning at the channel-wise group level to compress SAM's image encoder, coupled with knowledge distillation from intermediate layers to restore the pruned encoder. The red numbers highlight the pruned dimensions at each pruning step.

sub-structures, circumvents this issue. Whether pruning embedding or bottleneck dimensions, the intact remaining dimensions enable alignment through loss backpropagation. The effectiveness of this feature alignment, especially in data-scarce scenarios, highlights our framework's efficacy.

An overview of the alternate slimming framework is detailed in Figure 2. Given the Vit-based image encoder with k blocks, the output and intermediate features of each block within the encoder are denoted as $E = \{e_i\}_{i=1}^k$ and $H = \{h_i\}_{i=1}^k$. Specifically, for *Multi-Head Attention Blocks* (MHABs), the intermediate feature refers to the concatenated QKV features, while for the MLPs, it refers to the hidden features between two linear layers. The final output image embedding is represented as $t$. The original encoder is referred to as $v_0$, while the pruned encoders after embedding pruning and bottleneck pruning are denoted as $v_1$ and $v_2$, respectively. The alternate slimming process can be described as the following progressive procedure: embedding pruning, bottleneck aligning, bottleneck pruning, and embedding aligning.

**Embedding Pruning.** The embedding dimension significantly impacts the encoder's performance as it determines the width of features extracted within the encoder. To begin with, we prune the embedding dimensions $\mathcal{D}(E)$ while keeping the bottleneck dimensions $\mathcal{D}(H)$ constant. The presence of residual connections necessitates the preservation of uniformity in the pruned embedding dimensions $\mathcal{D}(\{e_i\}_{i=1}^K)$ across all blocks. Consequently, we employed uniform local pruning.

**Bottleneck Aligning.** In the context of incremental knowledge recovery, the pruned encoder learns from the original encoder's output $t_{v_0}$ and aligns with its dimensionality-consistent bottleneck features $H_{v_0}$ in each block. The distillation loss function for bottleneck aligning is a combination of bottleneck feature loss and final image embedding loss:

$$\mathcal{L}_{Bn} = \alpha \cdot \mathcal{L}_{MSE}(H_{v_0}, H_{v_1}) + (1-\alpha) \cdot \mathcal{L}_{MSE}(t_{v_0}, t_{v_1}), \tag{5}$$

where $\mathcal{L}_{MSE}(.,.)$ is mean-squared error, the dynamic weight $\alpha$ of $n$th epoch is defined as:

$$\alpha = \begin{cases} 0.5 & n < N \\ 0 & n >= N \end{cases}. \tag{6}$$

We set $N = 10$ for bottleneck aligning.

**Bottleneck Pruning.** Following the pruning of the embedding dimension $\mathcal{D}(E)$ and its coupled structures, we exclusively focus on pruning the bottleneck dimension. As the dimension of intermediate

features $\mathcal{D}(\{h_i\}_{i=1}^{K})$ in each block are entirely decoupled, we can systematically apply dimension pruning at various ratios for each block while maintaining the predetermined overall pruning ratio. This approach involves utilizing a global ranking of importance scores to conduct global structural pruning.

**Embedding Aligning.** The pruned encoder $v_2$ will learn from the embeddings $E_{v_1}$ and final image embedding $T_{v_1}$ from the pruned encoder $v_1$ to expedite knowledge recovery. Simultaneously, it also computes loss functions based on the final image embedding $t_{v_0}$ from the original encoder $v_0$ to enhance the precision of knowledge recovery. The total loss function for embedding aligning is defined as:

$$
\begin{aligned}
\mathcal{L}_{Emb} = \alpha \cdot (\mathcal{L}_{MSE}(E_{v_1}, E_{v_2}) + \mathcal{L}_{MSE}(t_{v_1}, t_{v_2})) \\
+ (1 - \alpha) \cdot \mathcal{L}_{MSE}(t_{v_0}, t_{v_2}),
\end{aligned}
\tag{7}
$$

where the dynamic weight $\alpha$ of $n$th epoch is defined as:

$$
\alpha = \begin{cases} \frac{N-n-1}{N} & n < N \\ 0 & n >= N \end{cases} .
\tag{8}
$$

The dynamic weight $\alpha$ will progressively diminish to zero as the distillation process unfolds. This transition in the learning objective of distillation gradually shifts from $v_1$ to $v_0$ contributing to a smoother knowledge recovery. We also set $N = 10$ for embedding aligning.

**Remark.** The implementation of alternate slimming on decoupled sub-structures significantly reduces the disruption to the original model, particularly when the pruning ratio is quite high. This strategy also preserves consistent dimensionality, enabling effective intermediate feature distillation, which is especially beneficial in data-scarce conditions. Consequently, in comparison to the previous pruning and distillation methods, our alternate slimming achieves a 3.40% and 0.92% increase in MIoU when the pruning ratios achieve 77% and 50%.

## 4 Experiments

### 4.1 Experimental Settings

**Implementation Details.** Our SlimSAM has been implemented in PyTorch [41] and trained on a single Nvidia Titan RTX GPU using only 0.1% (10,000 images) of the SA-1B [25] dataset. The base model of our framework is SAM-B [25]. The model's parameters were optimized through the ADAM [24] algorithm with a batch size of 4. Training settings for both bottleneck aligning and embedding aligning are identical. The pruned models undergo distillation with an initial learning rate of $1e^{-4}$, which will be reduced by half if validation performance does not improve for 4 consecutive epochs. The total training duration is 40 epochs for SlimSAM-50 (with a 50% pruning ratio) and 80 epochs for SlimSAM-77 (with a 77% pruning ratio). We exclusively compressed the image encoder while retaining SAM's original prompt encoder and mask decoder.

**Evaluation Details.** To ensure a fair quantitative evaluation of the compressed SAM models, we compute MIoU between the masks predicted by the model and the ground truth masks of the SA-1B dataset. We use the most challenging single-point prompts given in annotations for experiments. The results using box prompts are also reported in our Appendix. For efficiency evaluation, we provide information on parameter counts and MACs. Additionally, we present details about training data, training iteration and training GPUs for evaluating the training cost. Qualitative comparison of results obtained using point prompts, box prompts, and segment-everything prompts are also shown in the following section.

### 4.2 Comparision and Analysis

**Comparing with existing SAM compression methods.** As depicted in Table 1, we conducted a comprehensive comparison encompassing performance, efficiency, and training costs with other SOTA methods. Our SlimSAM-50 and SlimSAM-77 models achieve a remarkable parameter reduction to only 4.0% (26M) and 1.4% (9.1M)of the original count, while also significantly lowering computational demands to just 3.5% (98G) and 0.8% (23G) MACs, all while maintaining performance levels comparable to the original SAM-H. In contrast to other compressed models, our approach yields substantial performance enhancements while simultaneously achieving greater lightweight

Table 1: Comparing with other existing SAM compression methods on SA-1B dataset. We report parameter counts, MACs, training costs, and *Mean Intersection over Union* (MIoU) for a comprehensive and fair comparison.

| Method | Params↓ | MACs↓ | TrainSet | BatchSize | GPUs | Iters | MIoU↑ |
|---|---|---|---|---|---|---|---|
| SAM-H [25] | 641M | 2736G | 11M(100%) | 256 | 256 | 90k | 78.30% |
| SAM-L [25] | 312M | 1315G | 11M(100%) | 128 | 128 | 180k | 77.67% |
| SAM-B [25] | 93M | 372G | 11M(100%) | 128 | 128 | 180k | 73.37% |
| FastSAM-s [62] | 11M | 37G | 220k(2%) | 32 | 8 | 625K | 30.72% |
| FastSAM-x [62] | 68M | 330G | 220k(2%) | 32 | 8 | 625K | 35.41% |
| MobileSAM [60] | 9.8M | 40G | 100k(1%) | 8 | 1 | 100k | 62.73% |
| EfficientSAM-t [50] | 10M | 28G | 12.2M(110%) | 128 | 64 | 450k | 69.42% |
| EfficientSAM-s [50] | 26M | 94G | 12.2M(110%) | 128 | 64 | 450k | 71.19% |
| EdgeSAM [63] | 9.6M | 23G | 100k(1%) | 64 | 8 | 50k | 65.96% |
| **SlimSAM-50(Ours)** | 26M | 98G | **10k(0.1%)** | 4 | 1 | 100k | **72.33%** |
| **SlimSAM-77(Ours)** | **9.1M** | **23G** | **10k(0.1%)** | 4 | 1 | 200k | 67.40% |

Table 2: Comparing with other structural pruning methods. 'Ratio' signifies the pruning ratio applied to channel-wise groups. Training costs remain consistent for the same pruning ratio.

| Ratio | Method | Labels | Params↓ | MACs↓ | MIoU↑ |
|---|---|---|---|---|---|
| Ratio=0% | SAM-H [25] | ✔ | 641M | 2736G | 78.30% |
| | SAM-L [25] | ✔ | 312M | 1315G | 77.67% |
| | SAM-B [25] | ✔ | 93M | 372G | 73.37% |
| Ratio=50% | Scratch Distillation | ✘ | 26M | 98G | 1.63% |
| | Random Pruning | ✘ | | | 71.03% |
| | Magnitude Pruning [14] | ✘ | | | 69.96% |
| | Hessian Pruning [30] | ✔ | | | 71.01% |
| | Taylor Pruning [38] | ✔ | | | 71.15% |
| | **SlimSAM-50(Ours)** | ✘ | | | **72.33%** |
| Ratio=77% | Scratch Distillation | ✘ | 9.1M | 23G | 1.34% |
| | Random Pruning | ✘ | | | 62.58% |
| | Magnitude Pruning [14] | ✘ | | | 61.60% |
| | Hessian Pruning [30] | ✔ | | | 63.56% |
| | Taylor Pruning [38] | ✔ | | | 64.26% |
| | **SlimSAM-77(Ours)** | ✘ | | | **67.40%** |

and efficiency. SlimSAM consistently delivers more accurate and detailed segmentation results across various prompts, preserving SAM's robust segmentation capabilities to the greatest extent. This qualitative superiority over other models is visually evident in Figure 5 and 6. Our approach demonstrates outstanding levels of accuracy and correctness. Most notably, SlimSAM achieves these remarkable outcomes with exceptionally low training data requirements, utilizing merely 0.1% (10k) images of the SA-1B dataset. This represents a significant reduction in data dependency, requiring 10 times less data than both EdgeSAM and MobileSAM, and 1,100 times less data than EfficientSAM.

**Comparing with other structural pruning methods.** Having demonstrated structural pruning's efficacy for SAM compression, we established a benchmark for evaluating various pruning methods. SlimSAM is compared with four commonly used pruning methods: random pruning, magnitude pruning, Taylor pruning, and Hessian pruning, each employing different criteria for pruning. Additionally, we conducted comparisons with scratch-distilled models, which are randomly initialized networks sharing the same architecture as the pruned models. To ensure a completely equitable comparison, models with the same pruning ratios were subjected to identical training settings. Table 2 showcases our method's consistent superiority over other structural pruning techniques, particularly at higher pruning ratios. SlimSAM-50 and SlimSAM-77 outperform existing methods, achieving a minimum 1% and 3% MIoU improvement while incurring the same training cost. It is noteworthy that the performance of scratch distillation is extremely low at such a limited training cost. This further proves the effectiveness of our workflow in preserving knowledge from the original model.

## 5 Ablation Study and Analysis

We conducted a series of ablation experiments on the SlimSAM-77 model, which features an ambitious 77% pruning ratio. To ensure a fair comparison in the ablation experiments, all evaluated

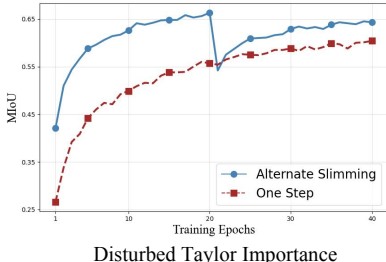
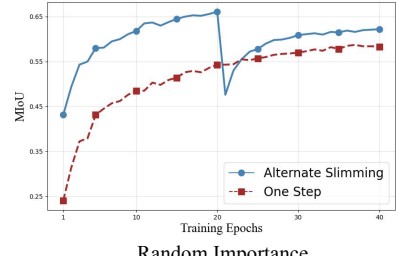

Disturbed Taylor Importance    Random Importance

Figure 3: Training results on SA-1B with the common one-step method and our alternate slimming framework. Left and right are results with disturbed Taylor importance and random importance.

Table 3: Comparison between disturbed Taylor pruning and original Taylor pruning.

| Method | MIoU↑ |
|---|---|
| Taylor Pruning | 62.04% |
| Disturbed Taylor Pruning | **62.31%** |
| SlimSAM-77 + Taylor | 63.63% |
| SlimSAM-77 + Disturbed Taylor | **64.48%** |

Table 4: Effect of distillation from intermediate layers and final output image embeddings.

| Step | Distillation Objective | MIoU↑ |
|---|---|---|
| Step 1 | Final Image Embeddings | 65.10% |
| Step 1 | + Bottleneck Features | **66.32%** |
| Step 2 | Final Image Embeddings | 63.91% |
| Step 2 | + Embedding Features | **64.48%** |

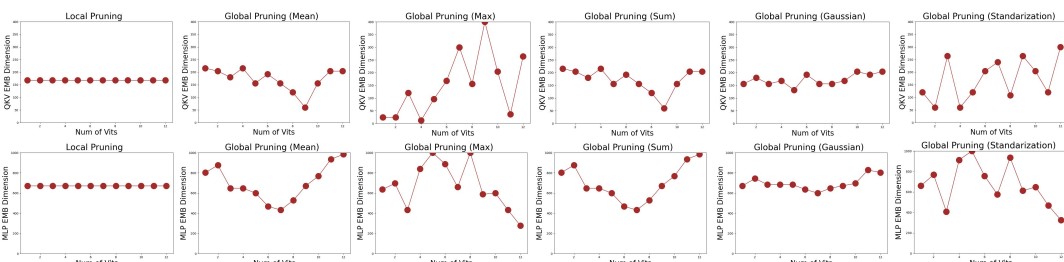

Figure 4: The intermediate dimensions of QVK Attention (top row) and MLP (bottom row) within each ViT after pruning. We present the outcomes of local pruning and global pruning under five distinct normalization methods.

models were trained for 40 epochs on the same 10k images from the SA-1B dataset. We also conduct additional experiments to evaluate the performance of SlimSAM with even less training data.

**Disturbed Taylor Pruning.** First, we conducted an ablation study to assess the impact of our proposed disturbed Taylor pruning on distillation. This innovative approach aligns the pruning criteria with the optimization objectives of subsequent distillation, resulting in improved performance recovery. As depicted in Table 3, our disturbed Taylor pruning consistently achieves significantly superior performance at the same training cost. For both the common one-step pruning strategy and our alternate slimming strategy, our method demonstrates MIoU improvements of 0.3% and 0.85% over the original Taylor pruning, respectively.

**Intermediate Aligning.** We also evaluate the effect of incorporating aligning with intermediate layers into the distillation process. As depicted in Table 4, distilling knowledge from intermediate layers leads to significant improvements in training results. Specifically, learning from bottleneck features and final image embeddings results in a 1.22% MIoU improvement for step 1 distillation, compared to learning solely from image embeddings. Similarly, for step 2 distillation, learning from embedding features and final image embeddings achieves a 0.57% MIoU improvement over the case where learning is based solely on image embeddings.

**Alternate Slimming.** In addition, we conducted experiments to investigate the impact of our alternate slimming framework. Unlike the common one-step pruning strategy, we partition the structural pruning process into two decoupled and progressive steps. In the first step, only the dimensions related to the embedding features are pruned, while in the second stage, only the dimensions related to the bottleneck features are pruned. Following both embedding and bottleneck pruning,

Table 5: Effect of global pruning evaluated under five different normalization approaches.

| Method | Normalization | MIoU↑ |
|---|---|---|
| Local Pruning | — | 64.38% |
| Global Pruning | Mean | 63.64% |
| | Max | 64.35% |
| | Sum | 63.55% |
| | Gaussian | **64.48%** |
| | Standardization | 64.14% |

Table 6: Comparision of training results using varied amounts of training data.

| Pruning Ratio | Data | Iters | MIoU↑ |
|---|---|---|---|
| Ratio=50% | 10k | 100k | **72.33%** |
| | 5k | 100k | 71.89% |
| | 2k | 100k | 69.79% |
| Ratio=77% | 10k | 200k | **67.40%** |
| | 5k | 200k | 64.47% |
| | 2k | 200k | 61.72% |

knowledge distillation with intermediate layer aligning is employed on the pruned model to recover its performance. For a more exhaustive analysis, we present the results obtained using different pruning criteria to assess whether the effectiveness of our method is influenced by importance estimation. As illustrated in Figure 3, our alternative slimming framework yields substantial improvements in MIoU, with gains of 3.9% and 3.5% observed under disturbed Taylor importance estimation and random importance estimation.

**Global Pruning vs Local Pruning.** Finally, we conducted experiments to evaluate the performance of local pruning and global pruning in bottleneck pruning. Given that the bottleneck dimensions in each block are entirely decoupled, we systematically applied channel-wise group pruning at various ratios for each block while preserving the predefined overall pruning ratio in this step. To obtain a consistent global ranking, we normalized the group importance scores $I_{\mathcal{G}}$ of each layer in five ways: (i) Sum: $I_{\mathcal{G}i} = \frac{I_{\mathcal{G}i}}{\sum_{i=1}^{K} I_{\mathcal{G}i}}$, (ii) Mean: $I_{\mathcal{G}i} = \frac{I_{\mathcal{G}i}}{\sum_{i=1}^{K} I_{\mathcal{G}i}/K}$, (iii) Max: $I_{\mathcal{G}i} = \frac{I_{\mathcal{G}i}}{Max_{i=1}^{K}(I_{\mathcal{G}i})}$, (iv) Standarization: $I_{\mathcal{G}i} = \frac{I_{\mathcal{G}i}-Max_{i=1}^{K}(I_{\mathcal{G}i})}{Max_{i=1}^{K}(I_{\mathcal{G}i})-Min_{i=1}^{K}(I_{\mathcal{G}i})+1e-8}$, (v) Gaussian: $I_{\mathcal{G}i} = \frac{I_{\mathcal{G}i}-\sum_{i=1}^{K} I_{\mathcal{G}i}/K}{\sigma_{i=1}^{K}(I_{\mathcal{G}i})+1e-8}$. As indicated in Table 5, local pruning ensures consistent performance, whereas global pruning raises the model's upper-performance limit. Global pruning's efficacy is highly dependent on the chosen importance normalization method. For our model, we opted for global pruning with Gaussian normalization, which yielded the best training results. Following global pruning, Figure 4 illustrates the dimensions of bottleneck features (QKV embeddings and MLP hidden embeddings) within each ViT in the image encoder. When applying mean, sum, or Gaussian normalization, the ViTs in the middle exhibit more group redundancy compared to those at the beginning and end. However, the pruned dimensions do not display distinct patterns when utilizing max or standardization normalization. The impact of global pruning becomes more pronounced with an increased number of training iterations. Specifically, when training extends to 80 epochs, the MIoU for global pruning exceeds that of local pruning by approximately 2%.

**Even less data.** As shown in Table 6, with a pruning ratio of 50%, a reduction in the volume of training data only marginally impacts the model's performance. Notably, even when trained with a limited dataset of just 2,000 images, our SlimSAM-50 model remarkably attains an MIoU of nearly 70%. However, as the pruning ratio is elevated to 77%, a decrease in training data more significantly affects performance. This leads to the inference that although our methodology, which integrates pruning and distillation techniques, mitigates the need for extensive training datasets, the availability of more training data can still enhance model performance, particularly at higher pruning rates.

# 6 Conclusion

In this paper, we present a novel data-efficient SAM compression method, SlimSAM, which achieves superior performance with minimal training data. The essence of our approach lies in the efficient reuse of pre-trained SAM, avoiding the need for extensive retraining. We introduce key designs to the compression method for enhancing knowledge retention from the original model in data-limited situations. Specifically, our alternate slimming framework carefully prunes and distills decoupled model structures in an alternating fashion, minimizing disruptions to the original model and enabling the intermediate feature alignment by consistent dimensionality. Furthermore, the proposed disturbed Taylor importance estimation rectifies the misalignment between pruning objectives and training targets, thus boosting post-distillation after pruning. SlimSAM convincingly demonstrates its superiority while imposing significantly lower training costs compared to any other existing methods.

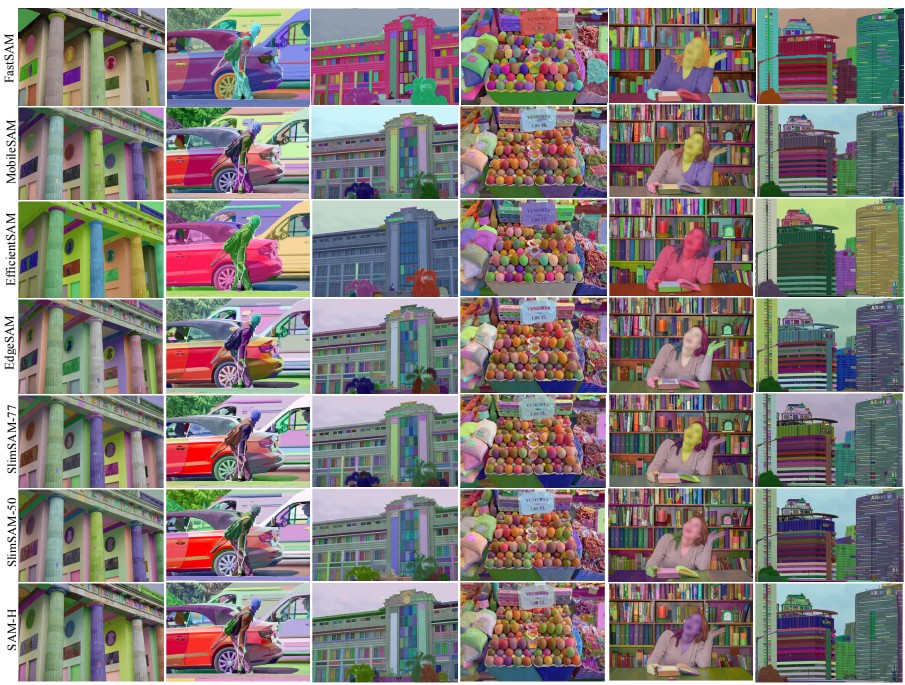

Figure 5: Comparison of segmentation results using segment everything prompts

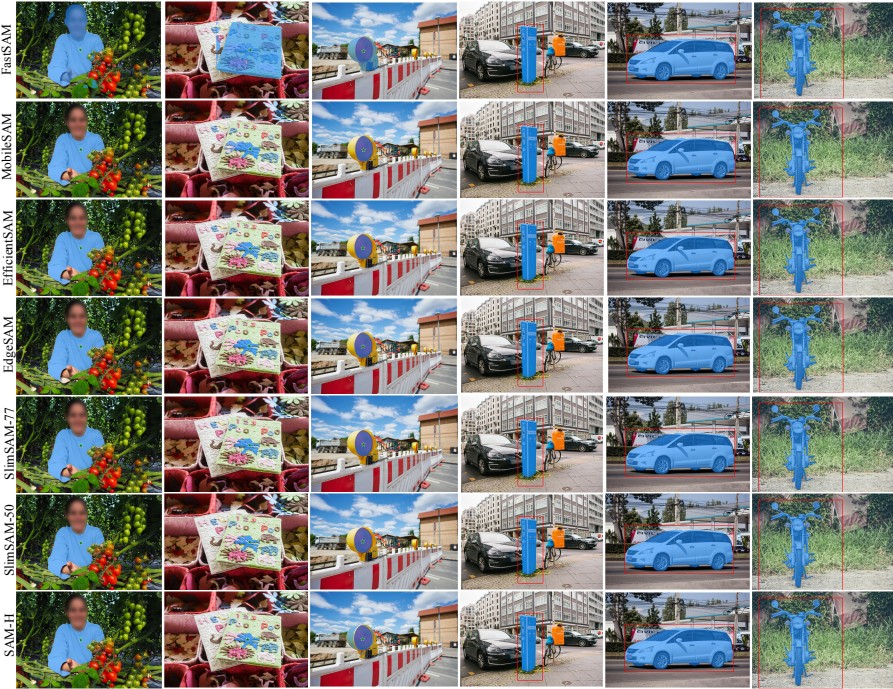

Figure 6: Left 3 columns: segmentation results obtained using point prompts; right 3 columns: segmentation results achieved with box prompts.

# Acknowledgement

This project is supported by the Ministry of Education, Singapore, under its Academic Research Fund Tier 2 (Award Number: MOE-T2EP20122-0006).

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

# A    Appendix

❊ In this document, we provide supplementary materials that extend beyond the scope of the main manuscript, constrained by space limitations. These additional materials include in-depth information about the ablation study of SlimSAM-50, further experiments assessing model efficiency, additional evaluations of training costs, analysis of dynamic loss, limitation discussion, and supplementary qualitative results.

## A.1    Ablation Study on SlimSAM-50

We performed an extensive series of ablation studies on the SA-1B [25] dataset utilizing the SlimSAM-50 model, characterized by its significant 50% pruning ratio. To guarantee a fair and consistent comparison across these ablation studies, each model under evaluation was uniformly trained over a span of 20 epochs, employing a training dataset comprising 10,000 images.

**Disturbed Taylor Pruning.** Initially, we executed an ablation study to evaluate the effects of our innovative disturbed Taylor pruning technique on distillation processes. This approach strategically aligns pruning criteria with the optimization goals of the ensuing distillation, thereby facilitating enhanced performance recovery. As illustrated in Table 7, our disturbed Taylor pruning method consistently outperforms, achieving markedly better results at equivalent training expenditures. In comparison to the conventional one-step pruning strategy and our alternative slimming approach, our methodology registers MIoU enhancements of 0.31% and 0.39% over the standard Taylor pruning method [38], respectively.

**Intermediate Aligning.** We further investigated the impact of integrating alignment with intermediate layers in the distillation process. As Table 8 illustrates, leveraging knowledge from these intermediate layers substantially enhances the training outcomes. Specifically, when distillation in step 1 incorporates learning from both bottleneck features and final image embeddings, there is a notable 1.19% improvement in MIoU compared to a methodology reliant solely on image embeddings. Similarly, in step 2 of the distillation process, a strategy that utilizes both embedding features and final image embeddings demonstrates a 0.29% MIoU improvement over approaches exclusively based on image embeddings.

Table 7: Comparison between disturbed Taylor pruning and original Taylor pruning.

| Method | MIoU↑ |
|---|---|
| Taylor Pruning | 70.02% |
| Disturbed Taylor Pruning | **70.33%** |
| SlimSAM-50 + Taylor | 70.42% |
| SlimSAM-50 + Disturbed Taylor | **70.81%** |

Table 8: Effect of distillation from intermediate layers and final output image embeddings.

| Step | Distillation Objective | MIoU↑ |
|---|---|---|
| Step 1 | Final Image Embeddings | 70.86% |
| Step 1 | + Bottleneck Features | **72.07%** |
| Step 2 | Final Image Embeddings | 70.52% |
| Step 2 | + Embedding Features | **70.81%** |

**Alternate Slimming.** Moreover, we conducted a series of experiments to examine the efficacy of our novel alternate slimming strategy. Diverging from the traditional one-step pruning approach, our method divides structural pruning into two distinct and progressive phases. In the initial phase, pruning is exclusively focused on the dimensions pertaining to embedding features. The subsequent stage then targets dimensions associated with bottleneck features. After completing both embedding and bottleneck pruning, we employ knowledge distillation with intermediate layer alignment on

Table 9: Effect of global pruning evaluated under five different normalization approaches.

| Pruning Method | Normlization | MIoU↑ |
|---|---|---|
| Local Pruning | — | 70.81% |
| Global Pruning | Mean | 70.76% |
| | Max | 70.77% |
| | Sum | 70.80% |
| | Gaussian | **70.83%** |
| | Standardization | 70.78% |

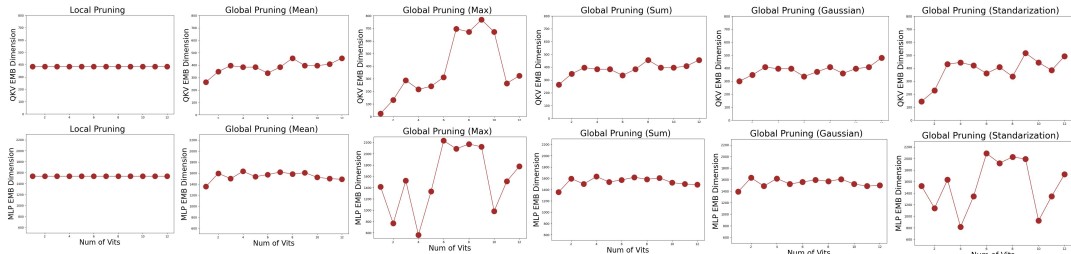

Figure 7: The intermediate dimensions of QKV Attention (top row) and MLP (bottom row) within each ViT after pruning. We present the outcomes of local pruning and global pruning under five distinct normalization methods.

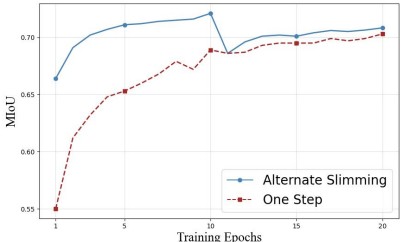

Figure 8: Training results on SA-1B with common one-step strategy and our alternate slimming strategy.

the pruned model to facilitate performance restoration. The outcomes derived from our proposed disturbed Taylor importance estimation are displayed in Figure 8. This figure demonstrates that our alternative slimming strategy significantly boosts MIoU, achieving an increase of 0.5%. When juxtaposed with the ablation study results of SlimSAM-77, it becomes evident that our strategy exhibits a more pronounced improvement, particularly when applied to models with higher pruning ratios.

**Global Pruning vs Local Pruning.** In our final set of experiments, we assessed the effectiveness of both local and global pruning approaches in the context of bottleneck pruning. Considering the complete decoupling of bottleneck dimensions in each block, we meticulously implemented channel-wise group pruning at varying ratios across different blocks. This was done while maintaining the predetermined overall pruning ratio for this phase of the study. To obtain a consistent global ranking, we normalized the group importance scores $I_{\mathcal{G}}$ of each layer in five ways: (i) Sum: $I_{\mathcal{G}i} = \frac{I_{\mathcal{G}i}}{\sum_{i=1}^{K} I_{\mathcal{G}i}}$, (ii) Mean: $I_{\mathcal{G}i} = \frac{I_{\mathcal{G}i}}{\sum_{i=1}^{K} I_{\mathcal{G}i}/K}$, (iii) Max: $I_{\mathcal{G}i} = \frac{I_{\mathcal{G}i}}{Max_{i=1}^{K}(I_{\mathcal{G}i})}$, (iv) Standarization: $I_{\mathcal{G}i} = \frac{I_{\mathcal{G}i} - Max_{i=1}^{K}(I_{\mathcal{G}i})}{Max_{i=1}^{K}(I_{\mathcal{G}i}) - Min_{i=1}^{K}(I_{\mathcal{G}i}) + 1e-8}$, (v) Gaussian: $I_{\mathcal{G}i} = \frac{I_{\mathcal{G}i} - \sum_{i=1}^{K} I_{\mathcal{G}i}/K}{\sigma_{i=1}^{K}(I_{\mathcal{G}i}) + 1e-8}$. Table 9 reveals that while local pruning maintains consistent performance, global pruning enhances the upper limit of the model's capabilities. In a departure from the findings observed with SlimSAM-77, various normalization methods do not markedly influence post-pruning performance. This suggests that the necessity of selecting an optimal normalization technique increases with the pruning ratio. For our model, we chose global pruning combined with Gaussian normalization, which led to the most favorable training outcomes. Figure 7 delineates the distribution of bottleneck feature dimensions (including QKV and MLP hidden embeddings) across each *Vision Transformer* (ViT) [6] in the image encoder. When mean, sum or Gaussian normalization is applied, the dimensions within the ViTs tend to distribute more evenly. However, employing max or standardization normalization often results in significant variances in the intermediate dimensions of each ViT.

## A.2  More Analysis on Efficiency

In the principal manuscript, the high efficiency of our SlimSAM model is objectively substantiated through the disclosure of parameter counts and Multiply-Accumulate Operations (MACs). This section extends the evaluation by reporting on actual acceleration in inference, further affirming the model's efficiency. As delineated in Table 10, SlimSAM-50 outperforms the original SAM-H

Table 10: Inference acceleration was empirically tested on an NVIDIA TITAN RTX GPU, revealing that higher pruning rates significantly improve inference speed.

| Pruning Ratio | Method | Speed Up↑ |
|---|---|---|
| Ratio=0% | SAM-H [25] | Faster×1.0 |
| | SAM-L [25] | Faster×1.7 |
| | SAM-B [25] | Faster×4.3 |
| Ratio=50% | SlimSAM-50(Ours) | Faster×6.9 |
| Ratio=77% | SlimSAM-77(Ours) | Faster×8.6 |

Table 11: The training results were compared using the varied amounts of training data but maintaining the same training iterations.

| Pruning Ratio | Training Data | Training Iters | MIoU↑ |
|---|---|---|---|
| Ratio=50% | 10k | 100k | 72.33% |
| | 5k | 100k | 71.89% |
| | 2k | 100k | 69.79% |
| Ratio=77% | 10k | 200k | 67.40% |
| | 5k | 200k | 64.47% |
| | 2k | 200k | 61.72% |

model by achieving a 6.9-fold increase in inference speed, while SlimSAM-77 attains an 8.3-fold acceleration. Our compression methodology markedly diminishes the actual inference time, concurrently effecting substantial reductions in both model size and MACs. The inference acceleration metrics were tested using an NVIDIA TITAN RTX GPU.

## A.3 More Analysis on Training Costs

In our foundational manuscripts, we demonstrate that SlimSAM exhibits exceptional compression performance with minimal training cost. A pertinent inquiry emerges: can SlimSAM maintain its competitive performance with reduced training costs? To address this, we have undertaken supplementary experiments focusing on the interplay between training cost and performance.

In Table 11, we present the results of additional experiments conducted with varying quantities of training data, while keeping the number of training iterations constant. We observe that with a pruning ratio of 50%, a reduction in the volume of training data only marginally impacts the model's performance. Notably, even when trained with a limited dataset of just 2,000 images, our SlimSAM-50 model remarkably attains an MIoU of nearly 70%. However, as the pruning ratio is elevated to 77%, a decrease in training data more significantly affects performance. This leads to the inference that although our methodology, which integrates pruning and distillation techniques, mitigates the need for extensive training datasets, the availability of more training data can still enhance model performance, particularly at higher pruning rates. It can be anticipated that with an increase in the volume of training data, our model may potentially achieve lossless compression of SAM.

Table 12 showcases the outcomes of experiments conducted by modifying the parameters for training iterations, while maintaining a constant training dataset size. The results clearly illustrate a direct relationship between the quantity of training iterations and the effectiveness of the model compression. It is evident that more extensive training significantly improves the performance of our compressed models. Remarkably, SlimSAM maintains its superiority over other methods even when the training iterations are halved, demonstrating its robustness and efficiency in achieving high-performance compression.

## A.4 More Analysis on Dynamic Loss

Further experiments were undertaken to assess the efficacy of employing dynamic loss within our intermediate feature alignment procedure. The outcomes of these ablation studies are detailed in Table 13. It was observed that a constant weight mechanism is more apt for scenarios involving a robust

Table 12: Training outcomes were evaluated using the same amount of training data across different numbers of training iterations.

| Pruning Ratio | Training Data | Training Iters | MIoU↑ |
|---|---|---|---|
| Ratio=50% | 10k | 50k | 70.83% |
| | 10k | 100k | 72.33% |
| Ratio=77% | 10k | 100k | 64.43% |
| | 10k | 200k | 67.40% |

Table 13: Ablation study on dynamic loss weights for distillation

| Step | Model | Constant ($\alpha = 0.5$) | Dynamic ($\alpha = \frac{N-n-1}{N}$) |
|---|---|---|---|
| 1 | SlimSAM-50 | **MIoU:72.07%** | MIoU:71.60% |
| | SlimSAM-77 | **MIoU:66.32%** | MIoU:65.79% |
| 2 | SlimSAM-50 | MIoU:70.42% | **MIoU:70.83%** |
| | SlimSAM-77 | MIoU:63.63% | **MIoU:64.48%** |

teacher model, whereas the implementation of a dynamic weight strategy enhances performance in instances where the teacher model exhibits lesser strength.

## A.5 Limitations

In this analysis, we critically examine the constraints of our methodology.

First, our approach demonstrates robust compression performance with minimal training data. Nonetheless, an expanded training dataset could further enhance the model's capabilities. Our current pre-trained SlimSAMs, limited by hardware constraints, are trained on a dataset of only 10,000 images from the SA-1B dataset. Utilizing a more comprehensive training dataset could potentially enable our method to achieve lossless compression.

Second, the essence of our method lies in employing structural pruning and knowledge distillation to preserve the knowledge of original pre-trained SAMs. This strategy inherently sets the performance ceiling of our model at the level of the original SAM. We found it challenging to surpass the performance of the original SAM, which acted as both the target for pruning and the target for optimization. A key area for future research will be exploring how to surpass the performance of the original SAM with limited parameter counts and reduced training costs.

## A.6 More Qualitative Results

We present more visual comparisons with other existing compressed models and the original SAM-H. Figure 9 provide a detailed visual comparison using the segment-everything prompt, while Figures 10 and 11 showcase additional qualitative results obtained with box prompts and point prompts, respectively. Relative to established compression models such as MobileSAM [60], FastSAM [62], EdgeSAM [63] and EfficientSAM [50], our model distinctly outperforms in achieving more precise segmentation, particularly noticeable at the object edges. Notably, even when benchmarked against SAM-H, our model demonstrates commensurate segmentation capabilities.

## A.7 Societal impacts

In this paper, we introduce SlimSAM, a novel data-efficient SAM compression method that delivers superior performance using minimal training data. SlimSAM achieves an outstanding compression ratio while preserving robust segmentation capabilities. This advancement enables the deployment of SAM on resource-constrained edge devices, underscoring its significant practical applications.

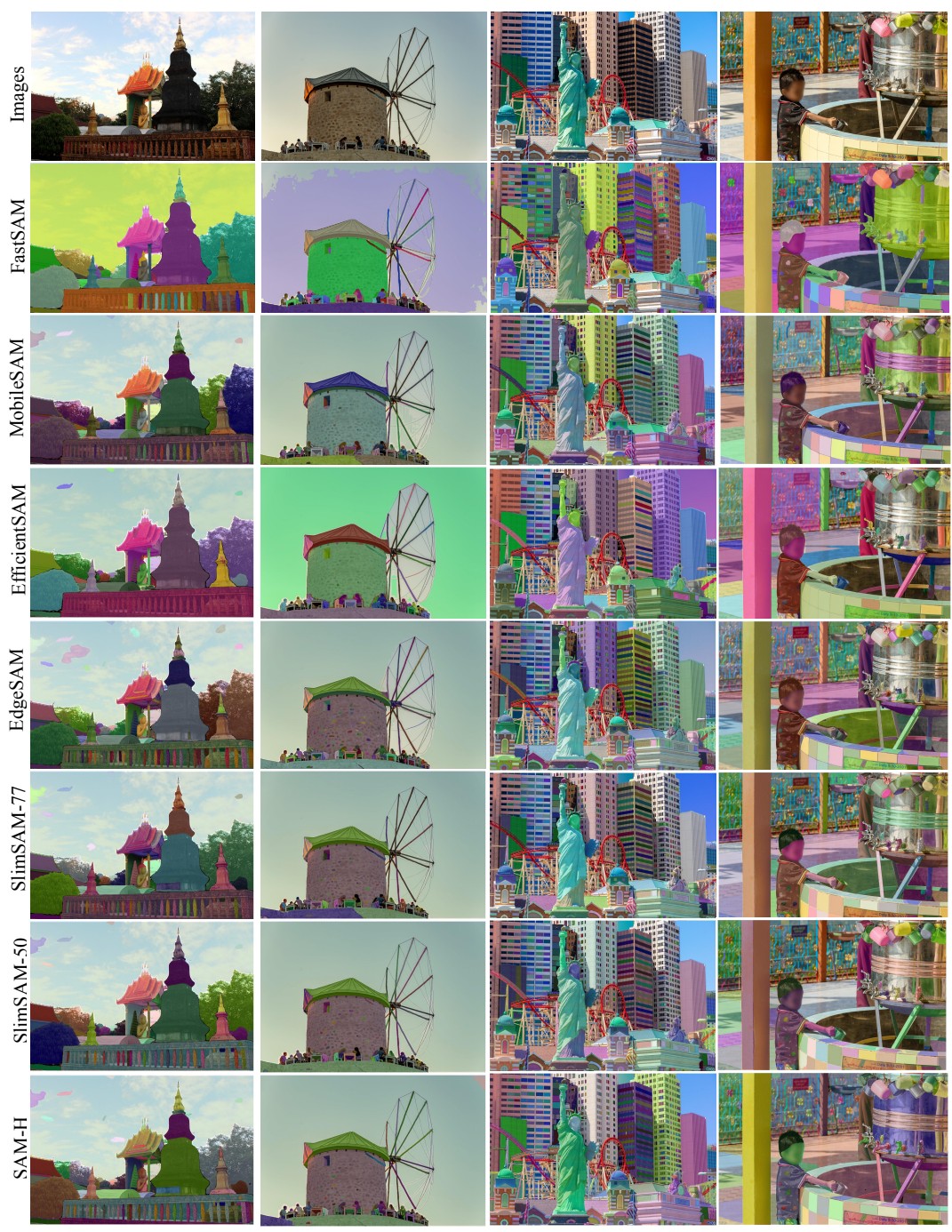

Figure 9: Comparison of segmentation results using segment everything prompts

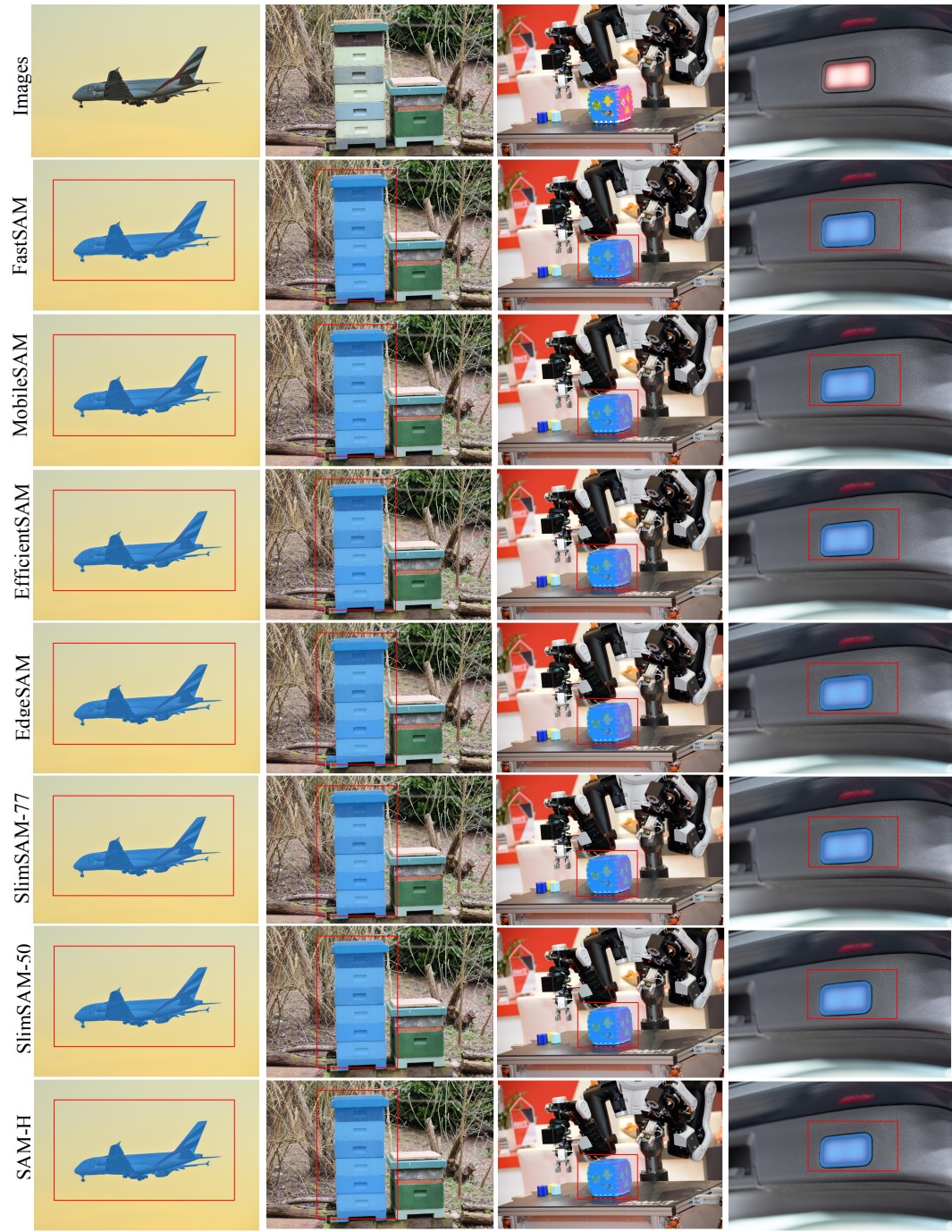

Figure 10: Comparison of segmentation results using box prompts

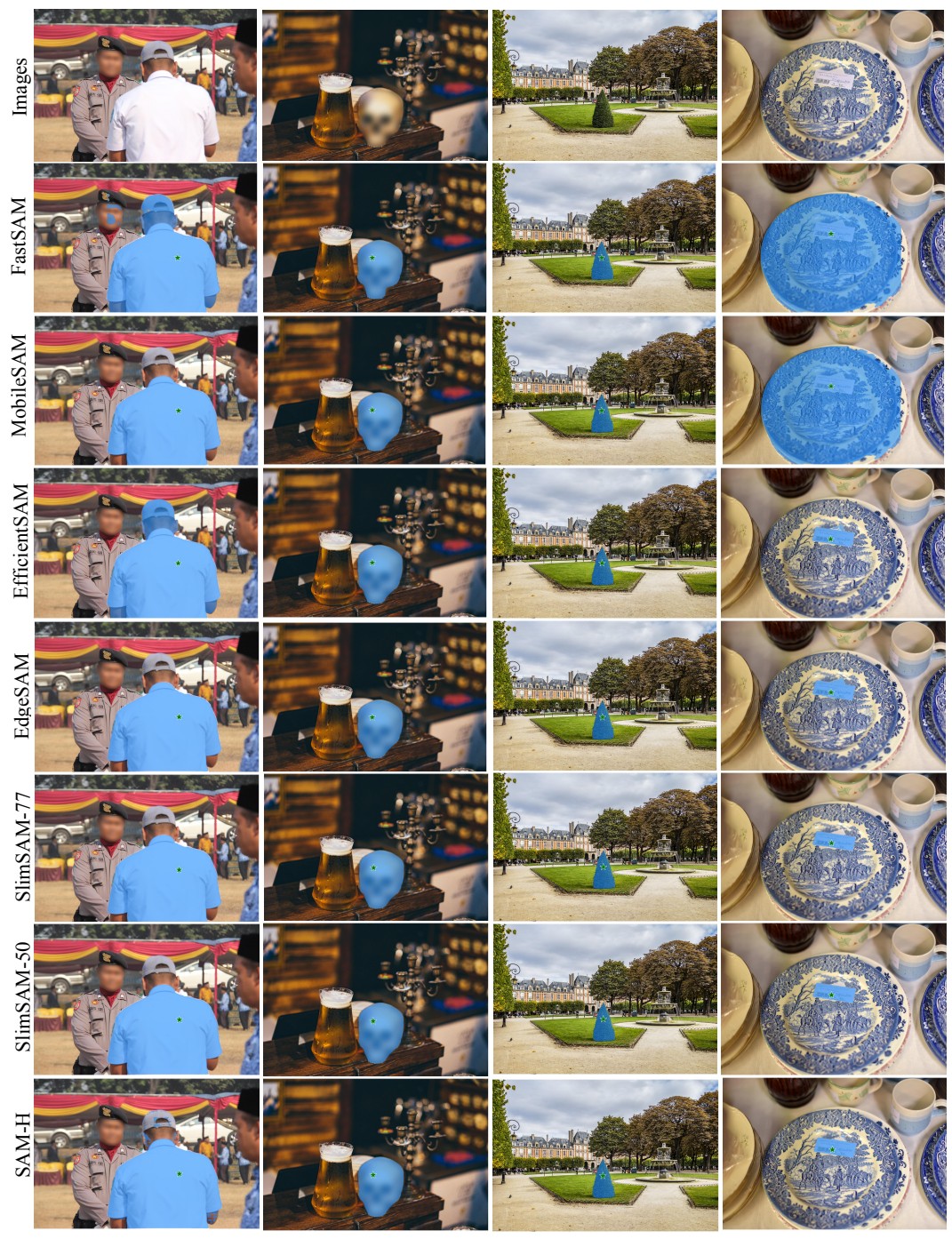

Figure 11: Comparison of segmentation results using point prompts

