# OpenReview forum: "SlimSAM: 0.1% Data Makes Segment Anything Slim"
_NeurIPS.cc/2024/Conference — NeurIPS 2024 poster_

### Official Review · Reviewer_5dUf · 2024-07-10

**Soundness:** 3
**Presentation:** 3
**Contribution:** 3
**Rating:** 5
**Confidence:** 5

**Summary:**

This paper introduces SlimSAM, a novel data-efficient method for compressing SAM. The authors propose an alternate slimming framework and disturbed Taylor pruning to enhance knowledge retention and compression performance with minimal training data. The method progressively prunes and distills distinct sub-structures, achieving significant performance improvements with only 0.1% of the original SAM training data. SlimSAM is validated through comprehensive experiments, showing superior performance and efficiency compared to existing SAM compression methods.

**Strengths:**

1. The alternate slimming method with dynamic alignment stratey seems interesting and reasonable.
2. Extensive experiments have been conducted to demonstrate the effectiveness of the proposed method.
3. The design of disturbed Taylor importance criterion is inspiring.

**Weaknesses:**

1. It would be better to analyze the impact of different Gaussian distributions on the pruning performance.
2. The impact of some hyperparameter settings on the pruning performance is under-explored.
3. Since the method combines both pruning and knowledge distillation techniques, it would be better to analyze the separate contributions of the two techniques.
4. Since the framework seems to be a progressive pruning manner and the pruning ratio of each dimension is a global preset value, I think splitting the two-stage pruning scheme into iterative pruning schemes would benefit the aggressive pruning, where the fine-tuning epochs and pruning ratio in each iteration are adjusted to make its compression cost equivalent to that of the two-stage pruning. So I am interested in the comparison of these two schemes.
5. There are several confusing symbol definitions, e.g., $N$ with two different usages in the paper.

**Questions:**

See weakness.

**Limitations:**

Not applicable.

---

> ### Author Rebuttal · Authors · 2024-08-06
>
> ## **Q1: It would be better to analyze the impact of different Gaussian distributions on the pruning performance.**
> Thanks for the valuable suggestions. The mean $\mu$ of Gaussian noise must be set to 0 so that the average of the sampled noise across the entire batch approximates zero. The Gaussian noise deviation $\sigma$ should be carefully chosen: a too-large deviation can cause the perturbed output to deviate significantly from the original distribution, while a too-small deviation can result in negligible gradients and inaccurate importance scores. Based on our experience, setting the deviation between 0.05 and 0.005 yields satisfactory pruning results.
>
> ## **Q2: The impact of some hyperparameter settings on the pruning performance is under-explored.**
> Thanks for the comments, this is indeed a question that needs further explanation and improvement.
>
> In our method, the **batch size** is set to 4, constrained by GPU memory (24GB). A larger batch size would result in a smoother training curve and better outcomes.
>
> The **learning rate** starts at 1e-4. If the model shows no performance improvement on the validation data for three consecutive epochs, the learning rate is halved.
>
> The total **training iterations** are 100k for SlimSAM-50 and 200k for SlimSAM-77. Increasing the iterations provides little improvement for SlimSAM-50 but can slightly enhance the performance of SlimSAM-77.
> ### Table 1 Effect of Different Training Iteration
> | **Pruning Ratio** | **Training Data** | **Training Iters** | **MIoU↑**   |
> |-----|-----|-----|-----|
> | Ratio=50% | 10k | 50k  | 70.83% |
> | Ratio=50% | 10k | 100k | 72.33% |
> | Ratio=77% | 10k | 100k | 64.43% |
> | Ratio=77% | 10k | 200k | 67.40% |
>
> Table 2 shows the effect of different **loss weight** settings during training. We use constant weights during stage 1 and dynamic weights during stage 2. Here, **n** represents the nth epoch, while **N** is set to half of the total training epochs. A smaller **N** leads to slower convergence, while a larger **N** results in worse performance.
> ### Table 2 Effect of Dynamic Weights Setting
> | **Step** | **Model**| **Constant (α=0.5)**| **Dynamic (α=(N-n-1)/N)**|
> |--------|--------|-------|------|
> | 1 | SlimSAM-50 | **MIoU: 72.07%** | MIoU: 71.60%|
> | 1 | SlimSAM-77 | **MIoU: 66.32%** | MIoU: 65.79%|
> | 2 | SlimSAM-50 | MIoU: 70.42%     | **MIoU: 70.83%**|
> | 2 | SlimSAM-77 | MIoU: 63.63%     | **MIoU: 64.48%**|
>
> ## **Q3: Since the method combines both pruning and knowledge distillation techniques, it would be better to analyze the separate contributions of the two techniques.**
> Thanks for the comments. Both pruning and distillation are indispensable in our method, as using them together is necessary for achieving satisfactory compression results.
>
> Without knowledge distillation, the final model performance is very poor (MIoU less than 30%) if the pruned model is trained with labeled data. This is because the complex coupling structure of SAM's image encoder and mask encoder requires a large amount of training data and batch size for effective training. This issue is also noted in the MobileSAM [1] paper.
>
> On the other hand, if we only use distillation without pruning and try to train a lightweight model with the same structure from scratch using such limited data (10k), the training results do not converge at all (MIoU less than 2%), as reported in Table 1 of our submission.
>
> [1] Zhang, Chaoning, et al. "Faster segment anything: Towards lightweight sam for mobile applications." arXiv preprint arXiv:2306.14289 (2023).
>
> ## **Q4: I think splitting the two-stage pruning scheme into iterative pruning schemes would benefit the aggressive pruning, where the fine-tuning epochs and pruning ratio in each iteration are adjusted to make its compression cost equivalent to that of the two-stage pruning. So I am interested in the comparison of these two schemes.**
> Thanks for the valuable comments. This is a quite interesting idea.
>
> We experimented with splitting the two-stage pruning of 50% of the dimensions into iterative pruning at 10%, 20%, 30%, 40%, and 50%. We observed two key findings:
>
> 1. With the same number of total training iterations, iterative pruning produces better results, especially when the number of iterations is low. However, this advantage gradually diminishes as the number of iterations increases.
>
> 2. Iterative pruning takes more training time when the number of total training iterations is the same.
>
> Carefully tuning the hyperparameters for iterative pruning might yield even better results, but this requires additional effort and expertise.
>
> ## **Q5: There are several confusing symbol definitions, e.g., N with two different usages in the paper..**
> Thanks for the careful review. We apologize for the oversight. We will correct this error and review the rest of our submission for any similar issues.

---

> > ### Comment · Reviewer_5dUf · 2024-08-13
> >
> > Thank you for your efforts! My concerns have been mostly addressed. I will retain my positive score.

---

> > > ### Author Response · Authors · 2024-08-13
> > >
> > > Thanks for your valuable feedback and time in reviewing our work, your insights are greatly appreciated.

---

### Official Review · Reviewer_2d2P · 2024-07-10

**Soundness:** 4
**Presentation:** 3
**Contribution:** 4
**Rating:** 8
**Confidence:** 4

**Summary:**

The authors present a data-efficient method for SAM 39 compression, SlimSAM, comprising of alternate slimming framework and the disturbed Taylor pruning. Comprehensive experiments are conducted, demonstrating the significant superior performance and data efficiency.

**Strengths:**

1. The paper is clearly written and well-organized.
2. The use of alternate slimming that combines pruning and knowledge distillation is innovative.
3. Comprehensive experiments are conducted.

**Weaknesses:**

1. There's a typo with $w_i$  on line 112.
2. The symbol $N$ is assigned different definitions on lines 111 and 187.
3.Curious about the significant performance drop in Figure 3—could it be related to the setting of N? Would this decline still occur if the epochs were increased to 40?
4. More experiments are needed to prove that 'SlimSAM preserves SAM’s robust segmentation capabilities'.
5. How does the model perform in terms of generalization? Could the authors conduct evaluations on additional datasets?

**Questions:**

As mentioned in Weaknesses.

---

> ### Author Rebuttal · Authors · 2024-08-06
>
> ## **Q1: There's a typo with $w_i$ on line 112.**
> Thanks for the careful review. We will correct this error in the next version and check the rest of the submission.
>
> ## **Q2: The symbol *N* is assigned different definitions on lines 111 and 187.**
> Thanks for the valuable feedback. We apologize for the oversight. We will clarify the definition in the next version.
>
> ## **Q3: Curious about the significant performance drop in Figure 3.**
> Thanks for the comment, this is indeed a question that needs further explanation.
>
> The significant performance drop in the process is due to model pruning. Our alternate slimming strategy involves a two-stage compression method. First, we prune the model's embedding dimension and align the bottleneck dimension through distillation from epoch 1 to epoch 20. After that, we prune the bottleneck dimension and align the embedding dimension through distillation from epoch 21 to epoch 40. The performance drop occurs after epoch 20 when the bottleneck pruning is implemented.
>
> ## **Q4: More experiments are needed to prove that 'SlimSAM preserves SAM’s robust segmentation capabilities'.**
> Thanks for the valuable feedback. To further demonstrate SlimSAM's robust segmentation capabilities, we thoroughly evaluated its zero-shot instance segmentation on the COCO2017 validation set. As shown in Table 1 below, SlimSAM maintains strong segmentation performance across small, medium, and large areas, indicating its ability to handle objects of various sizes. We also provide comprehensive qualitative results in the supplementary materials, showing that our model delivers segmentation results similar to the original model across different prompts and scenarios. Notably, SlimSAM achieves this using only 10k unlabeled images (0.1% of the SA-1B dataset). Increasing the amount of training data will yield even better results.
>
> ## **Q5: How does the model perform in terms of generalization? Could the authors conduct evaluations on additional datasets?**
> Thanks for the comments. To demonstrate the generalization of SlimSAM on other datasets, we evaluated its zero-shot instance segmentation on the COCO2017 validation set. As shown in Table 1, SlimSAM maintains a strong zero-shot capability very close to the original model, despite a high compression ratio.
>
> ### Table 1: Zero-shot Instance Segmentation on COCO
> | **Model**| **Params ↓** | **MACs ↓** |**COCO AP ↑**|**COCO AP ↑(small area)**|**COCO AP ↑(medium area)**|**COCO AP ↑(large area)**|
> |------|------|------|-------|------|------|-------|
> | SAM-B         | 93M  | 372G |43.4% |28.2%|47.4%|58.7%|
> | **SlimSAM-50**| 26M  | 98G  |42.8% |27.3%|46.8%|58.5%|
> | **SlimSAM-77**| 9.1M | 23G  |41.4% |26.2%|45.9%|57.4%|

---

### Official Review · Reviewer_YejR · 2024-07-11

**Soundness:** 2
**Presentation:** 2
**Contribution:** 2
**Rating:** 4
**Confidence:** 4

**Summary:**

This paper works on compressing the Segment Anything Model (SAM). The compression method has the advantage of using much fewer training data. It has an alternate slimming framework and the label-free importance estimation criterion, i.e., disturbed Taylor pruning. The alternate slimming framework minimizes the divergence from the original model and enables the intermediate feature alignment by consistent dimensionality.

**Strengths:**

SAM is an important type of image segementation model with many applications. The high computation costs associated with SAM is a huge challenge to be resovled. The paper works on faciliating the efficiency of SAM with fewer training data. It reduces the cost for the inference and pruning. The method is straightforward and not hard to follow.

**Weaknesses:**

1. Limited novelty. The pruning method simply follows the prediction errors introduced by prior works with an additional Guassian noise for the creation of disturbed image embedding. It's not clear why using the gradient of loss between the original image and Guassian-disturbed image embedding can serve as the importance evaluation metric. The slimming by reducing the divergence between the original and pruned models is also standard.

2. The motivation of the paper is not strong, or the writing of the paper can be improved. It seems to emphasize the data-efficienct compression technique, but the methods do not contain much space to say how this data efficiency is achieved. The benefit of using less training data is also not emphasized. If the goal of data efficiency is to reduce the computation cost to update the parameters of the pruned model, why not use the whole dataset but with fewer training epochs/iterations?

3. Limited evaluation. The paper only provides the results on SA-1B. How is the performance on other datasets? For instance, EdgeSAM provides results on SA-1B, COCO, and LVIS to evaluate both in-domain performance and zero-shot transferability. Can SlimSAM show better results acrooss different datasets?

**Questions:**

1. In Eq. (4), the paper introduces disturbed Taylor importance, with $\mathcal{N}$ as the Gaussian noise with zero mean and standard deviation $\sigma$.  How will the deviation values influence the performance and why 0.01 is a reasonable setting?

2. What is the rational of Eq. (4)? Why adding the Guassian noise can serve as the purpose of evaluating the importance of model weights?

3. How is the 0.1\% data sampled? Will different sampled sub-dataset influence the performance of the proposed method? If the data is randomly sampled, will better sampling methods such as K-means clustering to select representative data provide better results?

4. The paper tries to reduce the training cost for the pruned model to update the parameters with a subset of the training data. Why not use the whole dataset but with fewer training epochs/itertations, which also reduces the cost? Which way can lead to a better performance?

5. Can this method also be trained on whole dataset? With the entire dataset, will it result a better MIoU performance than the limited data case?

6. In Table 6, is the training epoch/iterations for different data amount the same?

**Limitations:**

Please refer to the weakness and questions.

---

> ### Author Rebuttal · Authors · 2024-08-06
>
> ## **Q1: In Eq.(4), the paper introduces disturbed Taylor importance, with the Gaussian noise with zero mean and standard deviation. How will the deviation values influence the performance and why 0.01 is a reasonable setting?**
>  The noise deviation should be carefully chosen: a too-large deviation can cause the perturbed output to deviate significantly from the original distribution, while a too-small deviation can result in negligible gradients and inaccurate importance scores. Based on our experience, setting the deviation between 0.05 and 0.005 yields satisfactory pruning results.
> ## **Q2: What is the rational of Eq. (4)? Why adding the Gaussian noise can serve as the purpose of evaluating the importance of model weights?**
> Thanks for the comment, this is indeed a question that needs further explanation.
>
> The main challenge we face is the misalignment between the pruning target and the distillation target when using hard labels to generate gradient-based importance scores. To address this, we need to use soft labels to compute the importance score. However, soft labels do not generate gradients by loss functions for Taylor importance estimation. To solve this, we added zero-mean Gaussian noise with a small deviation to the soft labels. The disturbed soft label creates sufficient gradients by loss function while keeping the slightly noisy output close to the original. When the batch size is large enough (>=100), the mean of the sampled Gaussian noise across the whole batch approximates zero. Now the parameter with the highest importance score is the one that causes the greatest deviation between the pruned model's output and the soft label.
>
> The experimental results in our submission show that this method will cause less damage to the model during pruning and does not rely on accurate hard labels.
> ## **Q3: How is the 0.1% data sampled? Will different sampled sub-datasets influence the performance of the proposed method? If the data is randomly sampled, will better sampling methods to select representative data provide better results?**
> Thanks for the valuable comment. The training set is a **random subset** of the SA-1B dataset, containing 11k images, with 10k used for training and 1k for validation. We trained using multiple different subsets under the same settings and found the difference in final results to be negligible (MIoU difference is less than 0.25%). As you said, using a better method to select the training data may provide better results, but this will lead to more costs and place higher demands on the total amount of data available.
> ## **Q4: Why not use the whole dataset but with fewer training epochs/iterations, which also reduces the cost? Which way can lead to a better performance?**
> Thanks for the valuable comment. Using the entire SA-1B dataset involves significant storage and download costs. It takes more than **15 days** to download under typical network conditions and requires around **20 TB** of storage space, which is impractical for many users. Additionally, some users prefer to compress and fine-tune SAM on their customized datasets, but such data is often limited and expensive. Our method is highly user-friendly, as it requires only 10k data samples and one 24GB GPU to complete the entire compression and fine-tuning process.
> ## **Q5: Can this method also be trained on whole dataset? With the entire dataset, will it result a better MIoU performance than the limited data case?**
> Thanks for the valuable comment. Our method can be trained on the whole dataset and will result in an even better performance. In Table 1, we present experimental results using 2k, 5k, 10k, and 20k training data samples. As shown in the table, the model's performance improves significantly with an increase in the amount of training data. However, as explained in Q4, increasing data brings more cost and is less user-friendly.
> ### Table 1: Comparision of Training Results Using Varied Amounts of Training Data
> *Note: We evaluated the effect of increasing data on SlimSAM-77.*
> | **Pruning Ratio** | **Training Data** | **Iterations** | **MIoU↑**   |
> |-----------|-----|------|---|
> | Ratio=77% | 20k | 200k | 68.10%  |
> | Ratio=77% | 10k | 200k | 67.40% |
> | Ratio=77% | 5k  | 200k | 64.47% |
> | Ratio=77% | 2k  | 200k | 61.72% |
> ## **Q6: In Table 6, is the training epoch/iterations for different data amount the same?**
> Yes, except for the training data, all other training settings are exactly the same for a fair comparison.
> ## **Q7: The paper only provides the results on SA-1B. How is the performance on other datasets?**
> Thanks for the valuable suggestion. To demonstrate the generalization of SlimSAM on other datasets, we evaluated its zero-shot instance segmentation on the COCO2017 validation set. As shown in Table 2, SlimSAM maintains a strong zero-shot capability very close to the original model. Additionally, our entire compression process only requires 10k unlabeled images (0.1% of the SA-1B dataset).
> ### Table 2: Zero-shot Instance Segmentation on COCO
> | **Model**| **Training Data** |**COCO AP**|
> |-----|-----|-----|
> | SAM-B  | 11M |43.4% |
> | FastSAM| 220k|37.9% |
> | EdgeSAM| 100k |41.9% |
> | MobileSAM| 100k |41.0% |
> | SlimSAM-50| **10k** |**42.8%**|
> | SlimSAM-77| **10k** |41.4% |
> ## **Q8: Difference between our method and standard technique**
> Thanks for the comment. The application of standard pruning methods falls short in achieving superior performance due to the unique challenges presented by SAM's coupled structure and our data-efficiency goal. To enhance performance, we propose an Alternate Slimming Strategy to minimize divergence from the original model and enable the intermediate feature alignment by consistent dimensionality which is very effective when data availability is limited. We also propose Disturbed Taylor Pruning to address the misalignment between pruning objectives and training targets. The comprehensive ablation study strongly proves the superiority of our new methods.

---

> > ### Comment · Reviewer_YejR · 2024-08-13
> >
> > I thank the authors for providing the rebuttal. After reading the authors' responses and the reviews from other reviewers, I still have concerns about 1) Limited novelty. The methods used such as prediction errors are standard and exist in other works [30]. The rationale behind adding Gaussian noise is still not well explained. I understand that the authors try to explain from the results aspect (without degradation). But I'm curious about why this works.  Do the authors assume that the soft labels should follow certain distributions or something else? 2) The presentations should be improved. The connection between data efficiency and the approach from current writing seems to be weak. 3) The motivation is not convincing. In the paper, the authors mention that the data efficiency can reduce the training cost. If so, better data sampling strategies such as methods used in many sparse training approaches can help reduce the training data and training cost. But in the rebuttal, the authors further mentioned "some users prefer to compress and fine-tune SAM on their customized datasets, but such data is often limited and expensive." Then, if this is the case, the paper should further study data sampling strategies that are not uniformly distributed as user data might be imbalanced.   Furthermore, if using the motivation of the original paper, pruning, and fine-tuning with the whole dataset but with fewer epochs/iterations might also be a meaningful baseline. The authors' rebuttal is that for the whole dataset, "It takes more than 15 days to download under typical network conditions and requires around 20 TB of storage space, which is impractical for many users." I thought the pruning only needed to be conducted once with the entire training dataset and the obtained model should be suitable for many users. While it's not necessary for each user to conduct individual training on the whole dataset, why the impracticability problem will exist is not clear.  3) The performance. In Table 2, even random pruning can reach an accuracy of 71.03\% with a sparsity ratio of 50\%. The proposed method seems to only beat random pruning by 1.3\%. Though the table mentions that the training costs remain the same for random pruning and the proposed method, I'm not sure how the training cost is measured. As random pruning does not need to calculate anything during pruning, I assume that at least the pruning phase with random pruning is more training cost-efficient.
> >
> > I would keep my current ratings for the paper.

---

> ### Author Response · Authors · 2024-08-14
>
> Thank you again for giving us the opportunity to further clarify. We offered additional explanation and look forward to your response.
>
> ## **(1) The methods used such as prediction errors are standard and exist in other works [30]. The rationale behind adding Gaussian noise is still not well explained.**
> Thanks for the insightful comment. The output of the backbone can not generate gradients using loss functions, since there is no ground truth for intermediate outputs. This makes gradient-based pruning methods not directly applicable in our case. Our key idea is to use the backbone outputs appropriately disturbed by noise to create sufficient gradients for importance estimation while keeping the slightly noisy output close to the original. The way we disturb the output is to add Gaussian noise with zero mean and small variance. However, as long as the noise has a zero mean and small variance, other types of noise can also be employed. This addition of noise generates enough gradients while keeping the overall distribution of the disturbed soft labels nearly identical to the original. This method is effective and enables the use of backbone outputs to compute gradient-based importance scores while keeping the pruning objective consistent with the distillation objective.
>
> In addition, our proposed alternate slimming strategy is another contribution. It minimizes divergence from the original model by decoupling the pruning process and enables intermediate feature alignment through consistent dimensionality, which proves to be highly effective, especially when data is limited. This is further supported by the optimization curve presented in Figure 3 of our submission.
>
> ## **(2) The connection between data efficiency and the approach from current writing seems to be weak.**
> Thanks for the valuable feedback, we will improve this in our next version. The key to achieving data efficiency lies in minimizing the disruption to the original model, which helps preserve its knowledge, and in enhancing the knowledge distillation process to more effectively restore that knowledge.
>
> 1. Better Preserving Knowledge: Our proposed disturbed Taylor pruning achieves more accurate importance estimation by aligning the pruning target and distillation target. Our alternate slimming decoupled the pruning to a progressive procedure further minimizing the disruption to the original model.
> 2. Better Restoring Knowledge: Compared to the common method, our alternate slimming enables the intermediate feature distillation by consistent dimensionality which is extremely effective when data availability is limited.
>
> ## **(3) The motivation is not convincing.**
> Thanks for the comment. **Our primary contribution lies not in the compressed model itself, but in the data-efficient compression method we have proposed.** This method has demonstrated the ability to achieve significantly better compression performance with far less data compared to other methods that require much larger datasets. As shown in Table 1, our approach can deliver performance with just **10k** images that other methods fail to achieve even with over **100k** images.
>
> For engineers looking to compress SAM on their limited private datasets, our method offers a significant advantage, enabling them to achieve superior performance compared to other methods using the same amount of private data. This is because our approach effectively preserves the key knowledge of the original model without the need for extensive data. Researchers can apply our method to compress their own SAM models according to their specific needs, and it can also be adapted to compress other ViT-based models. Our method not only delivers better performance in these tasks but also significantly reduces the data requirements. This makes it practical for many to compress SAM on their own, which we believe will be beneficial to the entire community.
>
> The impact of a better data sampling strategy is marginal when compared to our method. Our approach can achieve superior performance with only **10k** images, surpassing methods like FastSAM, MobileSAM, and EdgeSAM that require over **100k** images. Achieving such impressive results—**10x less data with better performance**—would be impossible using any data sampling techniques alone.
>
> ## **(4) The performance.**
> Thank you for the comment. We would like to provide further clarification. In the context of image segmentation tasks, a 1.3% MIoU difference is indeed not a small gap. It's also important to highlight that when the pruning rate reaches 77%, our method achieves a substantial improvement of around **5%** MIoU compared to random pruning.
>
> Regarding the training cost, we want to clarify that the mentioned "same training cost" refers specifically to the knowledge distillation phase. In fact, the cost of the pruning phase is negligible. **Whether using random pruning or our method, the pruning phase can be completed in just 20 seconds on a low-end CPU**.

---

### Official Review · Reviewer_wJAz · 2024-07-11

**Soundness:** 3
**Presentation:** 3
**Contribution:** 3
**Rating:** 7
**Confidence:** 3

**Summary:**

In this paper, the authors propose a novel and data-efficient SAM compression method called SlimSAM, which progressively compresses the SAM model by alternately pruning and distillation. They introduce disturbed Taylor pruning to address the misalignment between the pruning objective and the training target, ultimately achieving better performance with ten times less training data than other compression methods.

**Strengths:**

- The proposed method is highly efficient and reasonable, achieving SOTA performance among different compression methods.
- The experimental results are comprehensive.
- This paper is well-written and easy to follow.

**Weaknesses:**

- To the best of our knowledge, compressing the SAM model aims to make it more suitable for annotation scenarios in edge applications. However, the authors' experiments seem to compare only parameter count and computational cost, without testing the throughput of the compressed model in real-world scenarios. The reviewer hopes to see relevant discussions in the experiments to better demonstrate the method's practical value in real-world applications.
- The authors achieved impressive performance using only a small amount of data, which raises curiosity about whether this method can achieve further improvements with more data or if this method quickly reaches its performance bottleneck with more training data.

**Questions:**

See the weaknesses above.

**Limitations:**

Limitations are discussed.

---

> ### Author Rebuttal · Authors · 2024-08-06
>
> ## **Q1:the authors' experiments seem to compare only parameter count and computational cost, without testing the throughput of the compressed model in real-world scenarios. The reviewer hopes to see relevant discussions in the experiments to better demonstrate the method's practical value in real-world applications.**
> Thanks for the valuable comment. Table 1 shows the actual running latency and acceleration ratio. We compressed the large SAM model into an extremely smaller size, achieving a significant speedup while maintaining model capability. This makes SlimSAM ideal for deployment on resource-constrained devices in real applications.
>
> Additionally, our whole compression process only requires 10,000 unlabeled images (0.1% of the SA-1B dataset). This greatly reduces the need for training data, lowering the costs associated with storing and downloading large datasets. For users who need to compress and fine-tune SAM on their customized data, which is often limited and expensive, our method provides an effective solution.
>
> ### Table 1: The Evaluation of Real Running Speed.
> *Note: We measured the latency and speedup ratio of our SlimSAM on a TITAN RTX GPU. The latency refers to the time taken to infer a 1024x1024 image.*
> | **Pruning Ratio** | **Method**| **Latency ↓** |**Speed Up↑** |
> |------|------|-------|------|
> | Ratio=0%    | SAM-H  | 654ms |Faster×1.0|
> | Ratio=0%    | SAM-L  | 386ms |Faster×1.7|
> | Ratio=0%    | SAM-B  | 152ms |Faster×4.3|
> | Ratio=50%   | **SlimSAM-50 (Ours)**|94ms |Faster×6.9|
> | Ratio=77%   | **SlimSAM-77 (Ours)**|76ms |Faster×8.6|
>
> ## **Q2:The authors achieved impressive performance using only a small amount of data, which raises curiosity about whether this method can achieve further improvements with more data or if this method quickly reaches its performance bottleneck with more training data.**
> Thanks for the valuable comment. In Table 2, we present experimental results using 2k, 5k, 10k, and 20k training data samples. As shown in the table, the model's performance improves significantly with an increase in the amount of training data. In the future, we plan to use the entire SA-1B dataset to train a SlimSAM model and make it available to the community.
>
> ### Table 2: Comparision of Training Results Using Varied Amounts of Training Data
> *Note: We evaluated the effect of increasing data on SlimSAM-77.*
> | **Pruning Ratio** | **Training Data** | **Iterations** | **MIoU↑**   |
> |-----------|----------|-----------|-------------|
> | Ratio=77% | 20k | 200k | 68.10% |
> | Ratio=77% | 10k | 200k | 67.40% |
> | Ratio=77% | 5k  | 200k | 64.47% |
> | Ratio=77% | 2k  | 200k | 61.72% |

---

> ### Comment · Reviewer_wJAz · 2024-08-13
> **Official Comment by Reviewer wJAz**
>
> I would like to thank the authors for their rebuttal, and most of my concerns/questions are correctly answered. I decide to keep my score to 7 to accept this paper. And this score may be further raised after I consider other reviewer's comments.

---

> > ### Author Response · Authors · 2024-08-13
> >
> > Thank you for the insightful and constructive comment. Your expertise and time are greatly appreciated in helping to improve the quality of our work.

---

### Official Review · Reviewer_WP1s · 2024-07-12

**Soundness:** 3
**Presentation:** 3
**Contribution:** 3
**Rating:** 5
**Confidence:** 4

**Summary:**

previous works on SAM compression replace the heavy image encoder with lightweight counterparts which requires dealing with the trade-off between training cost and model performances. Thus the end performances are usually compromised. This paper proposes a sliming framework to perform pruning and knowledge distillation to maximally preserve the performances of the original SAM model while incur minimum training costs. Extensive experiments show that the proposes method is very effective compared to various baseline methods.

**Strengths:**

1. the paper proposes a novel method to compress SAM into significantly smaller one while greatly preserving its original performances. Extensive experiments show that the proposed method uses significantly few training examples and achieves comparable (or better) performances.

2. the paper is well-written and the experiments support the claims made in the paper

3. the ablation studies show more interesting findings such as even less data and different pruning strategies.

**Weaknesses:**

1. The final quality seems to highly depend on the selected training set, see my question below in the question section.

2. The results in table 1 are a little hard to read, it's better to separate them into methods that are based on different SAM models. Right now they are all mixed together, it may not be very meaning full to compare the performances across different base models.

3. the evaluation seems limited which focuses on SAM-B, it will be great to see how the method works on all SAM models.

**Questions:**

1. how is the training set (0.1%) selected? are there any impacts if a different training set is sampled and used to train SlimSAM?

2. since the training and evaluations are done on SA-1B, does the method generalize well to other datasets beyond SA-1B?

3. due to the structure of ViT, a strong uniformity has to be enforced. have the authors thought about ways to work around it so that the model could potentially be further compressed?

4. does the proposed method work on other ViT based models?

**Limitations:**

it will be great to see the proposed method work on larger SAM models.

---

> ### Author Rebuttal · Authors · 2024-08-06
>
> ## **Q1: how is the training set (0.1%) selected? are there any impacts if a different training set is sampled and used to train SlimSAM?**
> Thanks for the comment. The training set is a **random subset** of the SA-1B dataset, containing 11k images, with 10k used for training and 1k for validation. We trained using multiple different subsets under the same settings and found the difference in final results to be negligible (MIoU difference is less than 0.25%). Thus SlimSAM is not very sensitive to the selection of training set.
>
> ## **Q2: Since the training and evaluations are done on SA-1B, does the method generalize well to other datasets beyond SA-1B?**
> Thanks for the valuable comment. To demonstrate the generalization of SlimSAM on other datasets, we evaluated its zero-shot instance segmentation on the COCO2017 validation set. As shown in Table 1, SlimSAM maintains a strong zero-shot capability very close to the original model, despite a high compression ratio. Additionally, our entire compression process only requires 10k unlabeled images (0.1% of the SA-1B dataset).
>
> ### Table 1: Zero-shot Instance Segmentation on COCO
> | **Model**     | **Params ↓** | **MACs ↓** | **COCO AP ↑** |
> |---------------|------|------|--------|
> | SAM-B         | 93M  | 372G |43.4% |
> | **SlimSAM-50**| 26M  | 98G  |42.8% |
> | **SlimSAM-77**| 9.1M | 23G  |41.4% |
>
> ## **Q3: The results in Table 1 are a little hard to read, it's better to separate them into methods that are based on different SAM models.**
> Thanks for the valuable feedback, this is indeed a question that needs further explanation and improvement.
>
> Except for our method which is based on **SAM-B (93M)**, all other compression methods are based on **SAM-H (641M)**. Using such a large model as the teacher model means that they will need significantly more training costs than us in the knowledge distillation stage. On the contrary, our model is only based on the lighter SAM-B and uses only 10k unlabeled images as training data to achieve better compression performance than other methods, which once again proves the superiority of our method.
>
> ## **Q4: due to the structure of ViT, a strong uniformity has to be enforced. have the authors thought about ways to work around it so that the model could potentially be further compressed?**
> Thanks for the comment. As you mentioned, residual connections require the input and output dimensions of each ViT block to be uniform. However, because the intermediate feature dimensions in each ViT block are independent, we can apply dimension pruning at different ratios for each ViT block while keeping the overall pruning ratio consistent. In this work, we use a global ranking of importance scores to perform global structural pruning on the intermediate dimensions of each ViT block.
>
> Table 2 shows the intermediate dimensions of each ViT block after both local uniform pruning and our flexible global pruning. Our method breaks the uniformity enforcement by assigning different pruning ratios to different ViT blocks, resulting in a significant improvement in segmentation MIoU **(65.9%-->67.4%)**.
>
> ### Table 2: Intermediate Dimensions of Each ViT Block after Pruning
> *Note: We demonstrate the dimension comparison on SlimSAM-77.*
> | **Method**| **Type**|ViT[0]|ViT[1]|ViT[2]|ViT[3]|ViT[4]|ViT[5]|ViT[6]|ViT[7]|ViT[8]|ViT[9]|ViT[10]|ViT[11]|
> |----|----|----|----|----|----|----|----|----|----|----|----|----|----|
> | Uniform Pruning | Attention|168|168|168|168|168|168|168|168|168|168|168|168|
> | Uniform Pruning | MLP      |696|696|696|696|696|696|696|696|696|696|696|696|
> | **Flexible Pruning**| Attention|156|156|180|156|180|180|180|192|168|180|180|204|
> | **Flexible Pruning**| MLP      |756|768|696|732|732|648|624|648|684|744|744|756|
>
> ## **Q5: does the proposed method work on other ViT-based models?**
> Thanks for the comment. Our method is effective with any models that have numerous cascaded ViT blocks, especially in scenarios where data availability is limited. We are currently working on applying this compression scheme to other large ViT-based models and plan to make it available to the community in the future.
>
> ## **Q6: the evaluation seems limited which focuses on SAM-B, it will be great to see how the method works on all SAM models.**
> Thanks for the valuable suggestion. We are actively seeking additional hardware resources to support our work on compressing the 641M SAM-H model, and we plan to accomplish this in the future. It is a common observation in model compression that larger ViTs have greater parameter redundancy. Therefore, we believe that SlimSAM will yield even better results on SAM-H.

---

> > ### Comment · Reviewer_WP1s · 2024-08-13
> > **thanks authors**
> >
> > thank the authors for the rebuttal. my questions are resolved. After reading other reviewer's comments, I think although there is concern of limited novelty, the final performances are good. I remain my initial rating and recommend acceptance of this work. good luck!

---

> > > ### Author Response · Authors · 2024-08-13
> > > **Thanks Reviewers**
> > >
> > > We sincerely appreciate your dedicated time and effort in reviewing our submission. Your valuable feedback is greatly appreciated.

---

### Author Rebuttal · Authors · 2024-08-06

Dear Reviewers, Chairs,

We sincerely appreciate the time and effort you have spent evaluating our submission, and we look forward to the discussion stage. We will include the review stage results in the appendix of our next version.

---

### Author Response · Authors · 2024-08-10

Thank you again for the valuable feedback. We have submitted our rebuttal and hope that it addresses your concerns. If you have any further questions or need additional information, please feel free to contact us. We greatly appreciate the opportunity to clarify any points and look forward to your continued feedback.

---

### Decision · Program_Chairs · 2024-09-25

**Decision:**

Accept (poster)

**Comment:**

The submission introduces a novel method SlimSAM for compressing the Segment Anything Model (SAM) with an emphasis on data efficiency. SlimSAM employs an alternate slimming framework that combines progressive pruning and knowledge distillation, alongside a new disturbed Taylor pruning technique. This framework is designed to achieve competitive performance while using significantly less training data. Extensive experiments demonstrate that SlimSAM reduces parameter counts, MACs, and training data requirements by impressive margins while maintaining performance close to the original SAM.

The reviewers collectively appreciate the novelty and practical significance of SlimSAM, particularly its ability to achieve competitive performance with minimal training data. The evaluation and ablation studies are robust and provide compelling evidence to support the authors' claims. Furthermore, during the rebuttal phase, the authors effectively addressed concerns related to dataset generalization, real-world applicability, and the clarification of technical details. Given the strengths of this work and the authors' responses, SlimSAM makes a good contribution to the field of model compression and efficient segmentation models.